# Affirmative Action Policies in Higher Education in Brazil: Outcomes and Future Challenges

Rosana Heringer 

College of Education, Federal University of Rio de Janeiro (UFRJ), Rio de Janeiro 22290-902, RJ, Brazil; rosana.heringer@gmail.com

**Abstract:** This article presents the results of broad research developed by a group of Brazilian scholars to make an assessment of affirmative action policies in admissions to higher education in Brazil, namely the quota legislation in place since 2012, which reserves places for lower-income, Afro-Brazilians and indigenous students in federal universities. The research has been developed through a combination of methods including analysis of secondary data, case studies in six federal universities, and interviews with university leadership and faculty members. The article presents the main results of the study, including the effect of quota legislation in diversifying the profile of higher education students in the last decade. It shows that a significant proportion of Afro-Brazilian students have been able to access good-quality higher education because of this legislation. The article concludes by presenting some challenges faced by quota students in federal universities, especially related to student support policies and a sense of belonging. It also presents recommendations to improve these policies.

**Keywords:** affirmative action; Brazil; higher education; quotas; race relations; minority struggles; inequality



## 1. Introduction

Until recently, the adoption of affirmative action policies for black Brazilians was considered a distant dream. The black movement demanded specific policies. Occasionally, multinational companies raised concerns about the workforce in relation to their Brazilian branches. However, as recently as the 1990s, affirmative action seemed unlikely. In a few years, a major shift happened. The subject garnered a groundswell of attention in debates due to the preparatory process that led up to the landmark event in 2001, the World Conference Against Racism, Racial Discrimination, Xenophobia, and Related Intolerance. Beyond the mobilization of the black movement and the increasing visibility of their demands, an important aspect of this process has been public statements of some government entities, mainly the Instituto de Pesquisa Econômica Aplicada—IPEA (Institute for Research on Applied Economics), linked to the Ministry of Planning, Budget, and Management. The debate intensified during the World Conference, when the official report was released, including the recommendation for the adoption of quotas for black students in public universities (da Silva and Pereira 2013).

Since then, there have been major changes in the reactions of Brazilian society to proposals for the promotion of racial equality. Since 2002, social movements, NGOs, and academic institutions have produced studies of affirmative action in higher education. In 2002, not more than three public higher education institutions had adopted these policies. In 2003, the University of Brasília (UnB) was the first federal university to adopt affirmative action. Ten years later, the picture was very different. In 2011, 115 public institutions had enacted some kind of affirmative action policies for excluded groups (Jodas and Kawagami 2011; Heringer and Ferreira 2009).

With the creation of the Special Office for the Promotion of Racial Equality, President Lula's administration advanced the debate about how to increase black student access

to higher education. In July 2003, a working group was created to discuss this issue at the federal level. Initially, participants had varying perspectives about and levels of commitment to affirmative action. SEPPIR's leader, Minister Matilde Ribeiro, stated that the government supported the adoption of quotas since the presidential campaign in 2002, but it was necessary to intensify the discussion. In the same ceremony, the Minister of Education (MEC), Cristovam Buarque, stated that the working group did not have the creation of quotas as an objective.

Early in 2004, the MEC delivered a proposal for a presidential decree that called for authorizing federal universities to adopt quotas. The authors of the proposal attempted to legitimize this type of measure and avoid future judicial demands. The decree proposed the self-declaration of color or race as the mechanism for identifying the quota's beneficiaries and pointed to the need for minimum score attainment on the exams by the quota beneficiaries. After several days with no decision about which policy would be adopted, in January 2004 President Lula decided to send a bill to congress in order to elicit more debate on the issue. The new Minister of Education at the time, Tarso Genro, came up with a proposal for discussion a few weeks later: recruiting black, brown, and indigenous students, as well as students with special needs and former prisoners, to enroll at private higher education institutions and fill one hundred thousand vacant seats. This number would represent 25 percent of the seats available in private institutions. More than a third—37.5 percent—of private student slots were vacant. The government would provide tax exemptions to participating institutions (Heringer and Ferreira 2009).

Due to its controversial nature, this proposal generated broad discussion, evoking both criticisms and positive assessments. Without a doubt, the government redefined the debate and broadened the scope of the discussion, diluting the specific question of expansion of access to public universities. One of the criticisms received was exactly this: why not invest the equivalent amount of this tax break in the federal universities, expanding the number of places? The Ministry of Education responded that, even if this measure was taken, the number of places generated in the federal universities would be small. Although the proposal was still controversial, the government decided to create the University for All Program (Programa Universidade para Todos, PROUNI) through Decree 213 (2004), which later became Law 11,096 (2005). Between 2004 and 2011, the PROUNI program offered 1,128,718 scholarships, and 748,788 were used by students (Neves 2012).

An impressive number of federal universities have adopted quotas. The phenomenon has been studied by academics since 2000. A study by the research group NIREMA from the Catholic University of Rio de Janeiro (PUC-Rio) has shown that 57 public higher education institutions adopted affirmative action policies between 2005 and 2008 (Paiva 2013).

These policies were not adopted without polemics about their legitimacy and their accordance with the constitutional principle of equality. Throughout the past decade there were many articles published in the national press and intense debates in the academy about these policies, mainly those specific to Afro-Brazilians (Júnior 2010; Júnior and Campos 2013). The debates centered on how to identify who is black in Brazil and, therefore, could benefit from the policies. How higher education institutions would deal with students who were less prepared and might not perform well was also discussed.

In this context of controversy and criticism, some universities were sued because of the adoption of these policies. There was concern regarding the outcome of two cases considered by the Brazilian Supreme Court, both questioning the quota system. The Supreme Court's unanimous decision in 2012 in favor of the constitutionality of affirmative action policies, including racial quotas, was a major development. This result brought legal protection to the practices that had been implemented by dozens of institutions in the country, as well as legal, political, and ideological legitimacy to the viewpoint that affirmative action policies are fair and healthy for democracy. The detailed and decisive statements from Supreme Court members articulated the ideas and reflections formulated over many years by black and anti-racist activists. The legal recognition of affirmative

action policies in access to higher education in Brazil remains a historical landmark that will contribute to the consolidation of these policies in the country (Silverio 2012).

In 2012, congress approved new federal legislation that created mandatory affirmative action policies in all federal higher education institutions in Brazil (Law No.12,711, 29 August 2012). The implementation was planned over a span of 10 years and mandated 50 percent quotas in federal universities for public high school students, with specific racial sub-quotas according to the proportion of the Afro-Brazilian and indigenous population in each state.

In 2022, the Brazilian Black Movement has celebrated a decade of the most important affirmative action policy in the national context: the quota legislation (Law No. 12,711, of 29 August 2012) that made it mandatory for Brazilian federal universities to reserve a percentage of places to low-income, Afro-Brazilian and indigenous students (Brasil 2012).

Quotas are a type of affirmative action policy that aims to provide greater equality of opportunities for historically disadvantaged groups through actions that expand the insertion of these groups in the educational system, in the labor market, and in health services, among others (Hasenbalg 1977; Almeida 2019).

The quota law regulates affirmative action policies in access to federal public higher education with the aim of promoting slots into universities and federal institutions of secondary technical education. The law establishes, in each call, per course and shift, a minimum of 50% reservation of places for those who have attended all high school in public schools. Half of these places should be reserved for students with a family income of up to 1.5 minimum wage per capita.

Regardless of family income, the law provides a sub-quota for blacks, browns, and indigenous people, calculated in a proportion equal to the share of those in the population of each state according to the last demographic census. As of 2016, the law was amended to include a sub-quota for people with disabilities.

## 1.1. Brazilian Affirmative Action Policies in an International Context

The debate about the implementation of affirmative action policies in Brazil is not disconnected from the development of this discussion, proposals, and policy implementation at a global level. Although there is knowledge in Brazil about affirmative action programs in several different countries since the second half of the 20th century (Paiva 2013; Júnior and Zoninsein 2006; Moehlecke 2002), the most common experience of this type of policy that is quoted and more broadly known is the US experience (Medeiros 2013; Pires 2013; among others). The American experience has inspired the activism of the Brazilian Black Movement and has also served many times as a mirror to analyze the perspectives of racial equality policies in Brazil. As a consequence, many of the proposed policies to address racial inequalities in Brazil have been directly inspired by US affirmative action policies in the 1960s, 1970s, and 1980s.

Another important reference for the Brazilian Black Movement has been the debates and the advocacy work developed at the World Conference Against Racism in Durban, South Africa, in 2001 (Werneck 2009). That scenario has presented to Brazilian black leaders the need to develop concrete policies to address racial inequalities and the importance of making the Brazilian state responsible for the implementation of these policies.

At this point, the main concern of black movements was the issue of representation, understood as the need for more black and brown students' enrolment in Brazilian public universities. We can see the need for numerical representation as an important stage of the struggle for racial equality from the perspective of Fraser's argument of representation as a strategy for social justice in democratic societies (Fraser 2001). This perspective is relatively common when we look at strategies of underrepresented groups to increase their participation in spaces from which they have been historically excluded for such a long time. The Brazilian black and brown population has been excluded from public higher education for decades (Hasenbalg 1977; Paixão and Carvano 2008) and this has become one of the main attempts to transform this picture.

However, there has been a subsequent understanding from black activists in Brazil that the representation was only the first step. There were severe limitations to really producing a change in a context in which more black and brown students were enrolling in Brazilian universities, but these institutions were elitist, excluding, discriminatory; in one word, colonial. This is where the political process of affirmative action implementation in Brazil has a straight dialogue with critical diversity studies and de-colonial approaches to affirmative action (Bhambra et al. 2018). Works such as those of Icaza and Vazquez, about diversity in the University of Amsterdam, show very clearly the limits of demographic diversity: "While it is of utmost importance that universities reflect the demographic diversity of the societies they are supposed to serve, the question of demographic diversity falls short of addressing the question of decolonisation. How can the university address the role it has played in reproducing global inequalities?" (Icaza Garza and Vazquez 2018, p. 115).

In this sense, the most recent debates about the effective democratization and, we could say, decolonization of Brazilian public universities address issues like student support policies, changes in curricula, anti-discriminatory policies to face denounces of racism, and other types of discrimination, among other issues. There is a growing understanding that demographic representation is only a first step and that the task of transforming in a structural way the traditional (and colonial) universities is a long-term process.

*1.2. Affirmative Action in Brazilian Federal Universities*

The reservation of openings for specific groups in federal higher education institutions is part of a broader set of policies that seek to expand access to the university. Most of them result from the political struggle led by black organizations, indigenous peoples, people with disabilities, movements for the human right to education, and other social movements that historically denounce the elitist and white origin of Brazilian higher education (Gomes 2017; Heringer and Johnson 2015).

Such policies cannot be thought of in isolation, but rather in conjunction with other actions that include the expansion of vacancies in federal universities, the creation of new federal universities and institutes, new campuses, and new courses. Also, part of this set of measures is the University for All program (Prouni, the restructuring and expansion of the Federal Universities program (Reuni, the Unified Selection System (Sisu, and the adoption of the National High School Exam (Enem) as the main form of entry into Brazilian higher education (Klitzke 2018).

All these measures resulted in an increase in the number of enrollments in higher education, which reached 8.6 million in 2019, representing 21.4% of the Brazilian population aged 18 to 24 (INEP 2020). Most of these enrollments, however, took place in private higher education institutions. The continuous growth of enrollments in public higher education was interrupted in 2017 due to deep budget cuts, intensified by the approval of the "spending ceiling"[1] (EC 95/2016) and the de-structuring of the expansion programs of public universities after the institutional coup of 2016. These budget cuts have also contributed to the reduction in resources to student support policies that have been implemented in Brazil in a more structured way since 2010 when the federal government created the PNAES—National Student Assistance Program—with the following objectives: "I—democratize the conditions for young people to remain in federal public higher education; II—minimize the effects of social and regional inequalities on retention and completion of higher education; III—reduce retention and dropout rates; and IV—contribute to the promotion of social inclusion through education" (Brasil 2010).

In order to achieve these objectives, the PNAES defined actions that should be adopted in the following areas: housing; food; transport; health care; digital inclusion; culture; sports; early childhood education for students' children; and access, participation, and learning of students with disabilities (Brasil 2010). Although there was an increase in the public resources allocated to this program until 2016, a limited number of higher education students benefited from these program actions (Vargas and Heringer 2017).

Even with limits and insufficiencies, especially in relation to the need for more robust investment in student support policies, we recognize affirmative action policies as successful policies. Affirmative action policies, together with other policies designed to promote greater democratization of access to Brazilian higher education, have contributed decisively to "change the face" of the public university, making it closer to the portrait of the Brazilian population as a whole. These policies have required deeper transformations in the social function of the university in its forms of functioning, curricula, and research agendas. With this study, we conclude that these policies need to continue, to be deepened and improved, and to count, above all, with more public resources in a continuous and sustainable way.

In this context, the research: "Evaluation of affirmative action policies in higher education in Brazil: results and future challenges", whose results are presented here in a synthetic form, was carried out between March 2021 and June 2022, led by the Laboratory of Studies and Research in Higher Education of the Federal University of Rio de Janeiro (Lepes/UFRJ) and by the organization Educational Action (Ação Educativa). The research aimed to contribute to evaluating the effects and results of the quota policy implemented since 2012, including the following aspects:

(i)     To observe the effectiveness of legislation in diversifying the profile of higher education students in federal universities;
(ii)    Identify the success of institutions in the retention of quota students;
(iii)   Analyze graduation rates and identify the difficulties faced by institutions and quota students in their trajectory throughout higher education;
(iv)    Analyze other results of the quota system associated with proposals for changes in curricula and the development of antiracist initiatives within universities;
(v)     Document the expansion of black student activism, among other aspects;
(vi)    Analyze the level of institutionalization of quota law and affirmative action policies in universities.

From the perspective of institutionalization, the research also investigated how the institutional evaluation of Higher Education Institutions (HEIs), a dimension of Sinaes (National Higher Education Assessment), reflected the implementation process of monitoring and improvement of the quota policy. As a result of the research, a set of recommendations to society and, in particular, to policymakers aimed at strengthening affirmative action policies in public universities is presented.

The research project has brought together a group of Brazilian researchers, mostly linked to federal public universities, who have been working in the field of affirmative action policies for many years and are engaged in the fight against racism and race inequalities in higher education, to analyze the policy implementation and to show its results in different aspects, both for the benefited students and for society in general.

In this article, we will present the main results of each of the three axes of the research, as well as the main conclusions and some of the recommendations drawn from the findings and analyses of these results.

## 2. Materials and Methods

The research was developed between March 2021 and June 2022. The research team was composed of 39 researchers and research assistants distributed in different Brazilian states. All the research was conducted during the isolation period due to the COVID-19 pandemic, so the data collection, including interviews and focus groups, was conducted online. Despite these difficulties, the team was able to work in a very articulated way, discussing regularly major methodological decisions and splitting into smaller teams when necessary to develop a specific part of the work.

The main outline of the research design was defined by the senior coordinators at the early stage of the research project. Based on previous studies (Senkevics and Mello 2019; Silva 2020; Mello and Senkevics 2020; among others) there was an understanding that the secondary data collection of enrollments in federal universities during the first ten years of the quota law implementation was important, also to respond to a growing demand

of the Brazilian Black Movement. In different seminars and events, many black leaders have highlighted the importance of the availability of reliable data about the outcomes of ten years of the quota legislation. Moreover, a specific concern motivated the research: the fact that President Bolsonaro's government did not intend to develop an assessment of the affirmative action policies created in 2012. In more than one opportunity he has stated that did not agree with the quota system, especially using race criteria. At the same time, there was a political concern in 2021 that politically conservative congressmen might propose a review of the quota legislation in order to end it and not fix it. Because of all these reasons, the research project was relevant and necessary, with special attention to quantitative secondary data that could present the main outcomes of the program.

The other part of the methodological framework has been the definition of six case studies which would give a detailed analysis of the implementation of the quota law at an institutional level. Previous studies about specific universities have been developed, highlighting the local characteristics of each institution, internal debates, the demands from students, and the student support policies implemented in each context (Souza and Borges 2020; Reis 2017; Oliveira 2019; Nonato 2018; Klitzke 2018; Jesus 2019; Paiva 2013). However, each of these studies followed their own research questions, objectives, and methodology, making it difficult to establish any kind of comparison or integrated analysis between them. Therefore, the research senior coordinators have decided to conduct the selected case studies using the same research design as much as possible, including documentary research, interviews, and focus discussions with students. This has made it possible to analyze similar features in each university and gather similar types of information, with contributions to the whole analysis about affirmative action implementation. It was interesting to notice, for instance, that universities with different sizes in terms of student enrollment and located in different regions of the country presented similar difficulties in terms of the quota implementation.

From an epistemological standpoint, the researchers were very much aware of the challenges represented by collecting such an amount of data in different sets and from different sources. One of the main agreements about the methodological decisions that have been made were related to the need to not lose focus on the specific characteristics of each university and the regional and cultural dimensions in which they were involved.

At the same time, the research team that was dedicated to the analysis of the secondary data at a national level was committed to establishing a dialogue between the data analysis of this project and other studies that have been conducted using national data sets and showing relevant results about quota legislation at a national level. This dialogue was fruitful because it has shown that we already have a relevant group of researchers in Brazil dealing with these large sets of data and the importance of presenting this in an accessible way to larger audiences. This was one of the main concerns of this research project.

The third part of the research design was dedicated to an innovative analysis of whether and how affirmative action debate is present in self-assessment reports of federal universities. These reports are mandatory and have to be presented every year to the Ministry of Education. However, they are frequently prepared in a very bureaucratic way and do not circulate very much in the institution itself. Even with these limitations, these reports and official documents formally state the commitments of the federal universities with their goals and institutional planning. This is why we have chosen to analyze the approaches to affirmative action in these documents, both through the analysis of the reports themselves and through interviews with members of the institutional committee responsible for developing these reports in the universities. The findings drawn from these sources have been analyzed for the first time at a national level.

An important part of the research design was the literature review conducted by the researchers responsible for the three research axes. In the case of the quantitative studies, as mentioned above, the aim was to establish a dialogue with previous researchers on similar issues. In regard to the institutional assessment, there was the aim of learning from other analyses about the role of institutional self-assessment and its role in institutional planning

and outcomes. In relation to the case studies, all the senior researchers responsible for each of the universities' cases have extensive experience on the subject and have developed research about affirmative action implementation in recent years. Therefore, the aim of the literature review in this case has been to update the references based on the selected keyword so that the researchers could have a common basis to develop their analysis. The literature review was integrated into each case study, contributing in different ways to the comprehension of specific issues related to the subject. For instance, the researchers have found updated case studies about quota implementation or student support policies in their universities that they did not know about before the literature review.

Throughout the work and when the research results were presented to the Brazilian audience, the research senior coordinators faced many times questions about why there was such a variety of data collection and which were the challenges involved in analyzing all this amount of data. The main answer to this is the capacity that the research team has developed to deepen the analysis both at the national level and at the case study level, making it possible to have a very comprehensive analysis of the main outcomes of affirmative action policies in Brazilian federal universities. The researchers were able to present and discuss a wide range of information about these programs, including historical perspectives, quantitative results for each university, political controversies, student demands, and challenges ahead. I would take the risk to say that the result of the research project presented here in synthetic form allows the most comprehensive analysis of quota policies in Brazilian federal universities so far.

The project has Involved researchers from six federal universities, distributed in all regions of the country, for the development of a self-developed methodology that combines the following research axes (Figure 1):

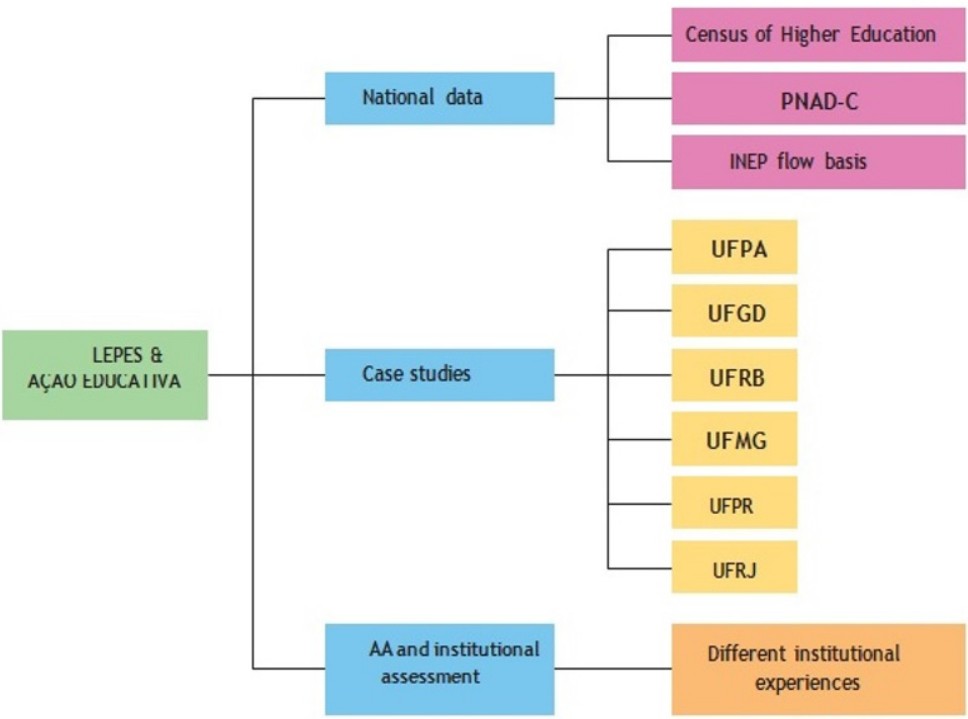

**Figure 1.** Research structure. Source: designed by the author, 2023.

Beyond the general coordination of researchers from LEPES and Ação Educativa, the selected six case studies were developed in the following federal public universities: UFPA (Pará Federal University); UFGD (Grande Dourados Federal University); UFRB (Reconcavo da Bahia Federal University); UFMG Minas Gerais Federal University); UFPR (Paraná Federal University); and UFRJ (Rio de Janeiro Federal University). The criteria for choosing these six universities were based on a previous assessment of the implementation of quotas

in the universities of each of the five Brazilian regions. Then, one university from each region was selected, reflecting the main characteristics of quota implementation in each region. Therefore, UFPA represents the North Region; UFGD represents the Center-West Region; UFRB represents the Northeast Region; UFMG represents the Southeast Region; and UFPR represents the South Region. UFRJ was included as a sixth case study because the researchers responsible for the study were based at UFRJ and have a specific interest in affirmative action implementation in this university. In addition to the development of a common theoretical framework and the critical analysis of the main results and challenges of the quota policy, the researchers responsible for each of the research axes developed the following activities.

*2.1. Axis 1: Secondary Data Collection at the National Level*

(i)     Review of previous studies;
(ii)    Systematization and analysis of Census of Higher Education data (2010–2019);
(iii)   Systematization and analysis of PNAD (National Household Survey) and PNAD-C (Continuous National Household Survey) data (2011–2019);
(iv)    Systematization and analysis of data from the INEP (National Education Research Institute, part of the Ministry of Education) flow base (entrants between 2010 and 2013; situation in 2017).

The analysis of secondary data at the national level has been divided into two parts: first, a bibliographical review of research on affirmative action in order to develop a theoretical framework that would enable the analysis of the main results of the quota policy; second, the creation of a database, bringing together quantitative information from the Ministry of Education (MEC) and the Brazilian Institute of Geography and Statistics (IBGE).

2.1.1. Bibliographical Review

The responsible team carried out a literature review that explored quantitative studies at federal universities with national coverage ranging from 2010 to 2021. This involved two procedures. On one hand, the academic production of scientific journals in the field of education was scrutinized through a general assessment of the works available on the Qualis platform. The terms "affirmative action" and "racial quotas" were searched in 121 scientific journals classified in the most prestigious journals and 380 in the second most prestigious group. Despite the abundant scientific production found in the research results, most articles are based on qualitative methods. Most of this research has focused on the analysis of specific institutional experiences of affirmative action implementation, showing senior management resistance to the new policy; lack of information from civil servants about how to implement the policy; and mistrust from faculty about the effectiveness of quota policies, among other results. However, as the focus here was quantitative studies, only ten articles were considered by the investigation.

The research results of the main actors involved in the related public debate were also analyzed, such as the National Association of Federal Rectors and the Multidisciplinary Affirmative Action Group. The 31 selected works share the intention of analyzing the results of the affirmative action policy in Brazilian higher education. The content analysis was carried out by the "quantitative team", focusing on issues such as methodology, use of data, and main results. The results here have shown the main changes in the access to higher education in the previous decade, with significant growth of lower-income, black, and brown students to public federal higher education. The data have also shown that, despite this growth, there is a structural racial and ethnic inequality that continues in Brazilian society, which has not yet been overcome by these recent quota policies[2]. More about these results will be shown later in this article.

The exercise of reading and analyzing the available bibliography was important to identify the main published results and also the main gaps, carrying out the systematization of data and the elaboration of a set of tables and graphs to compose the evaluation of

national data (both of the Brazilian population in their access to higher education, as well as of the student population at federal universities).

2.1.2. Secondary Data Processing and Analysis

The work with secondary data prioritized access to microdata to systematize information according to research interests, produced by two institutes: INEP and IBGE. The Anísio Teixeira National Institute of Educational Studies and Research (INEP) is a federal agency linked to the Ministry of Education (MEC) that periodically collects information about the Brazilian educational system. The data produced by this institute mobilized in this research were the *Higher Education Censuses* from 2010 to 2019. IBGE is a public institute of the Brazilian federal administration that generates information linked to geosciences, economic demographics, and social statistics. The data produced by this institute, used in this research, were the *National Household Sampling Surveys* (PNAD) and *National Continuous Household Sampling Survey* (PNAD-C) from 2011 to 2019. The *Higher Education Census* is conducted every year by INEP. The data from the survey are the most complete for understanding higher education institutions in the country, with information about undergraduate courses, students, and teachers. The collection comes from a registry known as the e-MEC System, which contains information from all institutions that offer higher education in the country. This record structure makes it possible to generate information about the infrastructure of the HEIs, vacancies offered, candidates, enrollments, freshmen, graduates, and teachers in different forms of academic organizations and administrative categories. The objective of the research was to generate valid and reliable data that allow knowledge about the Brazilian higher education system.

These data are the most reliable source for accessing information on the type of vacancy reservation requested by students since 2010. The data, therefore, allowed the monitoring of affirmative actions carried out between 2010 and 2019 based on variables that indicate, for example, whether the student entered the university by demanding a place reserved for public school graduates or a modality that combines reservation according to educational origin, family per capita income, and racial self-identification.

The PNAD and PNAD-C surveys are population sample surveys conducted by IBGE in a sample of Brazilian households researching different traits of society, such as population, education, and work. The data can be used to compute representative characteristics of the Brazilian population on an annual basis, except in census years. The microdata from these surveys contain important information for our research, such as people's racial self-identification, age, and educational level.

*2.2. Axis 2: Primary Data Collection through Six Case Studies in Selected Federal Universities*

(i)     Bibliographic survey on access and retention in each university studied;
(ii)    Documentary research;
(iii)   Interviews conducted with key informants (managers, course coordinators, professors, administrative staff) in each of the universities surveyed;
(iv)    Analysis of indicators of access and retention of quota students in each surveyed university, including selection of specific courses;
(v)     Holding conversation rounds with quota students and members of student academic collectives at each university.

Between July and December 2021, the majority of qualitative data was collected. Before starting data collection, the qualitative team discussed and worked together on important methodological definitions, reaching some decisions that we present below.

2.2.1. Bibliographic Research

The literature review focused on specific databases that provide important resources and reflections on the implementation of affirmative action policies in Brazil. These were the following data selected for research and distributed among the researchers:

- Working Group 11 (Higher Education) of the National Association for Research in Education (Anped);
- Working Group 21 (Education and Race Relations) of the National Association for Research in Education (Anped);
- Universitas database (Latin American regional platform with research results and other higher education resources in the region);
- Scielo (Scientific Electronic Library Online): Scielo is a bibliographic database, digital library, and cooperative model for the electronic publication of open-access periodicals. Scielo was created to meet the scientific communication needs of developing countries and offers an efficient way to increase visibility and access to scientific literature. The research selection included journals classified as A and B on the Qualis platform in the areas of Education and Social Sciences;
- National database of master's theses and doctoral dissertations (CAPES—Coordination for the Improvement of Higher Education Personnel);
- Qualis Platform (academic journals classified A and B in the areas of Education and Social Sciences by CAPES);
- ABPN (Brazilian Association of Black Researchers) Journal;
- UFRJ Minerva Database: database to search all resources available in UFRJ libraries.

  In all the databases listed above, the search was carried out using the following keywords:

1. higher education and quotas;
2. quota policy;
3. affirmative action and university;
4. black, brown, and indigenous students;
5. access and retention in HEIs.

The results of this bibliographical research were shared among the researchers and contributed to guiding the preparation of the interview script and conversation circles.

### 2.2.2. Documentary Research

We defined the collection of secondary data at each university researched, aiming to obtain more information about the process of implementing Law 12,711 (history; institutional documents; minutes of superior councils, among others). Within the defined scope, documents relating to affirmative action policies, access policies, and student support in each of the selected universities were researched and analyzed, including knowledge and analysis of how the university has dealt with committees to verify self-declaration of race. The availability of this documentation varied between different universities, with access predominating to documents available on institutional pages and also some that were made available by interviewees.

### 2.2.3. Courses to Be Analyzed at Each University

In order to delve a little deeper into the process of implementing the quota law at each university, we made a selection of courses to be analyzed in more depth in the case studies, based on the criteria of a course among the most selective, an intermediate course, and a course among the least selective.

The first step was to identify the cut-off grades in the year 2019 for all courses at the universities participating in the research. We researched the cut-off scores for all courses in the "Broad Competition", "Racial and Public School Quota", and "Racial, Income and Public School Quota" categories. Subsequently, we added and divided by three. This operation revealed the average of the (highest, lowest, and average) access scores for each institution. The formula for each operation is $(M1i + M2i + M3i)/3$, where M refers to the selected access modalities: 1. Broad competition, 2. Quota by Race and Public School, and 3. Quota by Race, Public School, and Income. i represents the variation between highest, average, and lowest grade obtained. It should be noted that the average values were obtained by calculating the simple arithmetic mean.

The identification of limit values allowed us to select a representative set of courses in the three modalities. However, in universities with a large range of courses, such as UFMG, UFRJ, and UFPR, the range of courses was still extensive. In the UFMG broad competition modality, for example, within the indicated ranges we identified 2 courses with the highest cut-off scores, 57 courses with intermediate cut-off scores, and 7 with the lowest cut-off scores. When reserving racial, income, and school origin places at UFMG, within the indicated ranges, we identified 1 course with the highest cut-off score, 29 courses with intermediate cut-off scores, and 4 with the lowest cut-off scores, which was still a high number of courses.

Therefore, we sought to verify the recurrence of courses in the different access modalities and classification bands. From this smaller group of courses, we decided that the person responsible for the case study at each university would define the course according to the area of knowledge, among those selected in the table, to achieve a greater diversity of student profiles.

After completing this selection, data collection related to these three courses at each university took place through three main strategies: interviews with the coordinators of these courses; invitation to quota students on these courses to participate in conversation circles; and analysis of secondary data made available from specific crossings of data from the historical series from 2010 to 2019 of the Higher Education Census.

### 2.2.4. Interviews

Interviews were carried out with managers of the universities selected for case studies, raising their perceptions and critical assessment of the process of expansion and democratization of higher education in the last ten years on the policies of access, retention, and academic success of students. Initially, the following managers were defined to carry out interviews: Dean; Vice-Rector; Pro-Rector of Undergraduate Studies; Dean of Student Policies (or equivalent); unit directors and/or coordinators of the three courses selected at each university; coordinators of the Hetero-identification Committee (or equivalent)[3]; and Coordinator of the Afro-Brazilian Studies Center (or equivalent). The possibility of including other important institutional actors in the discussion of these themes was also left open. This pre-selection would give an approximate total of 10 interviews at each university, totaling around 60 interviews for the set of case studies.

After defining the initial group of interviewees, there were some difficulties in terms of scheduling to carry out all the planned interviews. Thus, the number of interviewees at each university was unequal, leading to a total of 49 interviewees across the 6 universities, including deans, department chairpersons, and faculty members, among others.

In the interviews carried out, a collectively constructed question guide was used, with a set of questions organized into different blocks, with some blocks in common, addressing the identification of the interviewee and aspects of knowledge of the quota law, and other blocks directed to specific questions involving the implementation of quotas, based on the institutional position of each interviewee.

### 2.2.5. Organization of Conversation Circles with Students

The researchers responsible for the case studies organized and led conversation circles with students from each university. As a general guideline, the following composition of the groups was defined:

- A conversation circle with quota students, preferably from the three courses selected for analysis at each university, considering variations in gender, color, ethnicity, length of time on the course, and, as far as possible, those who did not form part of any group. The intention was to understand their conceptions about the university experience as quota students.
- A conversation circle was initiated with students who make up different student groups at each university (women, black students, indigenous students, LGBTQIA+ students, and so on)—starting from the perspective of intersectionality and possible

identity overlaps—which contribute to the perception of the complexity of human and social research. The aim was to understand to what extent student groups were enunciating agendas resulting from the incorporation of new profiles of students enrolled through affirmative action (Guimarães et al. 2020).

As the field research progressed, we considered the difficulties in mobilizing students and holding groups in the context of remote teaching, and in the final months of the year, we adjusted the composition and format of conversation circles at some universities. Even so, we tried to follow a common script, in an effort to establish a dialogue that would provide spaces for listening and exchange between students in the most horizontal way possible.

### 2.3. Axis 3: Study on the Role of Institutional Evaluation of SINAES in the Implementation of the Law

(i)    Bibliographic survey on the place of quota law and affirmative action policies in the field of higher education assessment;
(ii)   Documentary research;
(iii)  Interviews with members of the Self-Evaluation Commissions (CPAs), provided for in the National Higher Education Assessment System (SINAES);
(iv)   Analysis of institutional evaluation reports of universities sent to INEP.

The team began its work by carrying out a bibliographic survey on institutional assessment, affirmative action, and quota legislation. The research was carried out in the following databases: Scielo; ABPN Journal; Journal of Educational Assessment Studies; and Essays: Educational Assessment and Policies Magazine. Then, the abstracts were read by the team and articles were selected for analysis.

The team also analyzed the Institutional Development Plans (PDI) and reports from the Permanent Assessment Committees (CPA) of the six universities defined for case studies, in the following years: 2012–2014–2016–2018–2020.

In June 2021, a workshop "Institutional Assessment in the Higher Education Assessment System (SINAES) and the Implementation of the Quota Law" was held, in which 17 researchers from different backgrounds participated. We count on the important contribution of several members of the National Forum of Rectors of Community Engagement and Affirmative Actions of Brazilian Federal Universities (Fonaprace). The Institutional Assessment workshop aimed to generate subsidies for refining the methodological design and questions to be addressed by the research.

The team conducted interviews with the coordination and at least one other member of the CPA at the six universities. The team also conducted interviews with leaders from two selected universities to generate different perspectives on the use of institutional assessment for the implementation and evaluation of quota legislation.

It is important to mention that the research proposal was submitted to CONEP (National Council for Research Ethics) and analyzed by the Research Ethics Committee of the Center for Philosophy and Human Sciences (CFCH) at UFRJ. After analysis, the research was approved under the Certificate of Presentation of Ethical Appreciation (CAAE) number: 56179021.5.0000.5582.

### 3. Results and Discussion

#### 3.1. Analysis of Secondary Data: The Change in Students' Profiles

The quota law and other affirmative actions implemented in higher education have effectively contributed to democratizing access to federal universities and brought about a profound transformation in the references, meanings, and priorities of universities (research agenda, curricula, changes in procedures and institutional cultures), with a power to induce change in other public universities and private institutions in the country.

Not only are quota students benefiting from these policies, but the Brazilian higher education system as a whole benefits from the greater racial and social democratization of higher education, with new subjects, knowledge, provocations, experiences, and civilizing values calling into question the colonial project that marks Brazilian society to this day.

Racial quotas have been decisive for the access of black and indigenous students, demonstrating that social quotas alone are not enough to democratize this population's access to higher education (Senkevics and Mello 2019). The presence of PPIs (Black, Brown, and Indigenous) became more visible in the most selective courses, in which, before the quotas, they were practically absent. One way to demonstrate this numeric change along the time span analyzed in the research is the increase in students who have entered federal universities through the different types of social and racial quotas, as shown in Figure 2.

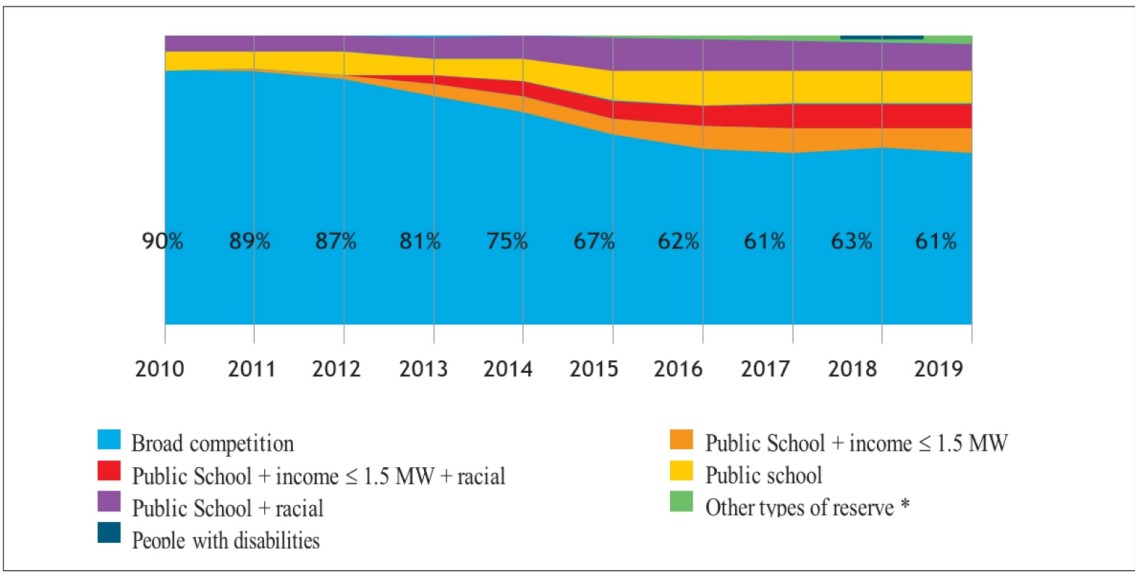

Source: INEP, Census of Higher Education. Preparation: Educational Action and LEPES UFRJ.

\* *Other types of reserve* refer to the vacancies of the institutions' own programs.

**Figure 2.** Entrants in present courses at federal universities by type of vacancy used—Brazil, 2010–2019 (%).

As presented in Figure 2, in 2010 the majority of entrants were through broad competition, that is to say, not considering any of the social and racial criteria. Since then, the proportion of students who benefited from social and racial quotas has increased year after year, reaching 39% in 2019. Among those that entered through quotas, a significant proportion used the racial criteria, which is combined with public high school and family income.

It is worth mentioning that Brazilian quota legislation operates through the combination of these different characteristics. This was a compromise solution adopted when the bill was discussed in Brazilian congress, along more than ten years. The first bill, in the late 1990s, proposed only the racial quota as criteria, following the discussions of the Brazilian Black Movement. However, due to the resistance to the use of racial criteria, other congress people have come up with this combination of race, income, and attending public high school, making it easier to get the approval in a politically conservative congress (H. Santos et al. 2013). This has been the main form of affirmative action adopted in Brazilian universities in the last 20 years.

The controversies, fears, and resistance that marked the initial moments of the public debate on affirmative actions in higher education throughout the 2000s diminished, the reality of the policy imposed itself, and different actors adapted to its existence. However, despite the greater politicization of the public debate on racism in Brazilian society in the last decade, thanks to the actions of the Black Movement, the understanding that "social quotas are enough" can still be identified in the discourse of certain university managers and course coordinators (Heringer and Carreira 2022). Our research indicates

the opposite: social quotas alone are not enough to tackle the immense racial inequality in higher education.

Among students enrolled in federal universities, those who had the highest increase between 2010 and 2019 were those who entered places reserved for graduates of public schools, black, brown, and indigenous (and people with disabilities), combined with the income criterion (Figure 3). This can be interpreted as a relevant result in order to promote greater inclusion and more democratic access to Brazilian federal universities because the combination of these criteria expresses the condition of those students who have been the most marginalized and discriminated against in their path to access higher education. For a long time in Brazilian history, since the first higher education institutions were created in the early 19th century, lower-income, black, brown, and indigenous students were underrepresented in these spaces (Hasenbalg 1977; Fernandes [1972] 2007). This gives the dimension of the transformations Brazilian federal universities went through after the quota law.

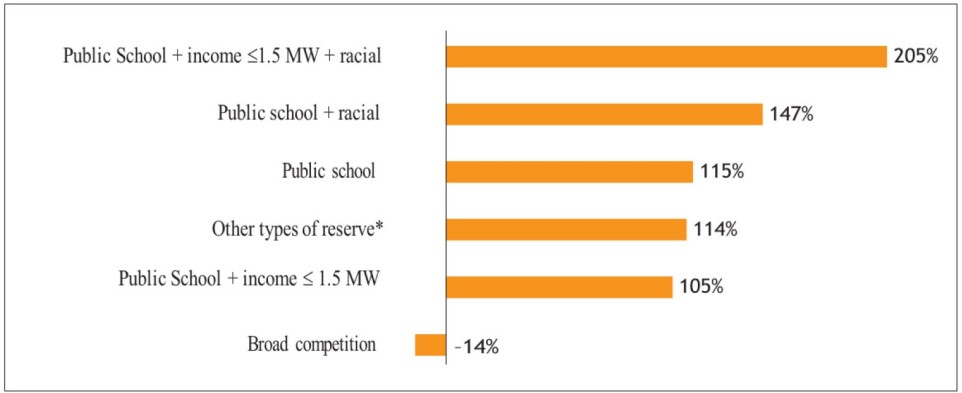

Source: INEP, Census of Higher Education. Preparation: Educational Action and LEPES UFRJ.
\* *Other types of reserve* refer to the vacancies of the institutions' own programs.

**Figure 3.** Percentage variation in the number of entrants by reservation of places, Federal Universities—Brazil, 2013–2019.

Although there were significant improvements in the enrolment of lower-income, black, brown, and indigenous students in Brazilian federal universities, the size of the structural social and racial inequalities in Brazilian society still has to be strongly addressed. Analyzing secondary national data available, our research has shown that the racial gap in higher education enrolment persists in the proportion of one black student for every three white students. It is observed that, in the black contingent, self-declared browns are proportionally greater in number in the group of graduates than black students (Figure 4). These data have been collected through the National Household Survey, investigating the school attainment of the whole population. This gives a broader picture of the scenario over the decade, showing that there is much more space for the growth of black and brown admissions to higher education beyond the current quota legislation.

The quota policy does not only aim to promote access to higher education. The evaluation also depends on the guarantee of students' conditions of retention that enable successful trajectories within the university environment. One way to verify this is to document the path of students and calculate whether the dropout and completion rates of students entering via the reserving policy are similar to those of students entering the general competition. It was possible to follow the generation that entered the federal higher education system in 2013 after the quota law. The dropout rates after the first year between quota and non-quota students were very similar: 11% and 10%, respectively. In varied courses such as architecture and urbanism, civil engineering, electrical engineering, medicine, veterinary, and pedagogy, the rates among entrants in 2013 are equally close (Honorato et al. 2022).

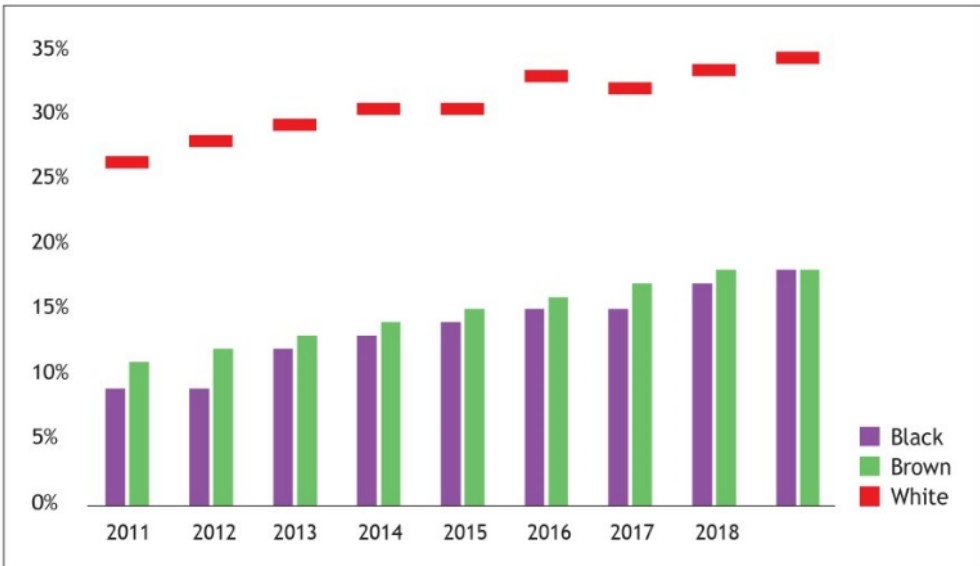

Source: IBGE, Pnad and Pnad-C 2019. Preparation: Educational Action and LEPES UFRJ.

**Figure 4.** Net undergraduate attendance rate by color/race in Brazil, 2011–2019.

These results corroborate previous research that indicated that, although they come from relatively less privileged socioeconomic situations, quota students have educational results very similar to those of non-quota students. This is a strong signal that students recognize and value the opportunities they achieve.

However, beyond the advancements in the last decade, the research has shown that the institutionalization of quota policies across federal universities is still limited, as they are generally understood to be the responsibility only of bodies and instances linked to student support or responsible for the agenda of promoting ethnic–racial equality. Most universities fail to embrace the quota law and other affirmative actions as structural and transversal policies, as will be shown in the next section.

### 3.2. Institutional Changes and Student Support Policies

The results presented in this section were generated from the six case studies conducted in the five regions of Brazil. Each of them gives examples of implementation of the quota legislation in different institutional and local contexts, for example, bigger and smaller HEIs, metropolitan and non-metropolitan contexts, and regions with greater and smaller proportions of black, brown, and indigenous populations. These different characteristics add specific forms of implementing the quota legislation, as well as student support policies that can contribute to student retention and, more specifically, quota students' retention in higher education.

One of the first obstacles that the research team has faced has been the limited production and analysis of information about the path and demands of quota students in the universities that were researched. Knowledge about their trajectories was still disjointed, with limited institutional appropriation for planning purposes. In general, there is little systematized information available about the profile of quota holders, their academic trajectory, their performance, their difficulties and challenges, their demands, and their potential. We concluded that this is due to the difficulties of the universities in collecting and systematizing data about the students in general, but also to the lack of priority given to the production of information about quota students. For a long time, Brazilian federal universities were used to receiving students from more privileged backgrounds and were not very concerned about the eventual difficulties the students might have in their academic trajectories. It is relatively common to hear from university faculty that in the past students would arrive at the university "ready", with good academic records, not many economic difficulties, and no great problems of adapting to the university as a new space. This was a

statement that was heard many times during our research. In comparison to the reality of students after the quota law, teachers state that this situation has changed and students now are not ready but need greater support to stay, adapt, and perform well in the university. This type of perception raises many reflections about the role of the university in receiving and contributing to the development of all the students, and not only those that were considered "ready".

We highlight the importance of federal universities expanding their commitment to the retention and academic success of all students, particularly those who enter through reserved places (Coulon 2008). Despite efforts, universities still do not have good programs for monitoring the performance and progress of quota students, facing difficulties related to the full acceptance and participation of these students in the institution. There is little knowledge on the part of some course coordinators about the difficulties faced by quota students, including those of an academic nature.

The profile of these students provoked a greater demand for student support policies in a broader perspective, involving financial aid, food, housing, transportation, and mental health programs (Vargas and Heringer 2016), in a context of resource cuts at federal universities, resulting in fiscal adjustment policies, anchored in Constitutional Amendment 95/2016. Known as the "spending ceiling", EC 95 established resource cuts for social and environmental policies for 20 years, while at the same time not setting limits for the drain of public resources into the financial system.

As a result of this situation, the volume of public resources allocated to the National Student Assistance Program (PNAES) has been much smaller than the growing demand for access to the benefit, intensified in the context of the economic and pandemic crises that have deeply affected quota students and their families, generating dissatisfaction and legitimate demands from students towards university bodies (Vargas and Heringer 2016; Heringer et al. 2022). Often, the Deans of Student Policies (or equivalent) tend to concentrate most of their efforts on the conflictive management of insufficient financial benefits and end up paying less attention to the pedagogical dimension and the daily lives of students.

Created in 2010, the PNAES urgently needs to be strengthened and transformed into a public policy with a legal basis approved by the National Congress with a stable source of resources, which supports the demands of the democratization process of federal higher education institutions, amplified by the Quota Law.

Our research has also shown that student groups, self-named as "student collectives" (black, indigenous, *quilombolas* (people from traditional black rural communities), LGBTQIA+, feminists, people with disabilities, etc.) have played a fundamental role in the retention of quota students in universities and in maintaining the memory and legacy of anti-racist political struggle promoted by black and indigenous movements and which is at the origin of affirmative action policies. There were different statements from students in the researched universities showing that these collectives were spaces of solidarity, common trust, exchange of information, study groups, and so on (Heringer et al. 2022).

The research also investigated the actions developed in terms of pedagogical support and integration of quota students into universities (Honorato and Heringer 2015). There is a long way to go: universities do not have a good program to monitor the performance and trajectory of these students. In some of the universities surveyed, there is an institutional body aimed at monitoring the academic trajectory of black, brown, and indigenous students, in order to identify difficulties and propose support, but in many cases, these bodies have few human resources and suffer from fragile institutional conditions. The research has concluded that it is important that public universities begin to have a greater variety of student profiles, with different trajectories, experiences, and information about the universe of higher education. Such students demand scholarships and financial aid, but also demand welcome, feedback, listening, broad information about academic opportunities, and dialogue with different instances of the university.

*3.3. Affirmative Action and Institutional Evaluation*

A component of the research project has been the inductive power of the institutional assessment of Higher Education Institutions (HEIs), provided for by the SINAES—National Higher Education Assessment System in the implementation of the quota law and affirmative action policies.

Institutional assessment implies the evaluation of the educational institution in all its dimensions: the curriculum, the relationships, and attitudes in the daily life of the institutions, the infrastructure, the valorization of education professionals, the conditions of access and retention of students, teaching and evaluation practices, democratic management, facing educational inequalities, and valorization of diversity in the educational environment. According to the proposed indicators and methodology, participatory institutional evaluation strongly contributes to a process of collective formation, construction of diagnoses, transformative action plans, and profound changes in institutional cultures.

In higher education, institutional assessment is one of the basic components of SINAES, established in 2004, and is developed in two main instances: self-assessment- carried out by the institution's own assessment commissions (CPAs), and external assessment, carried out by external commissions designated by the National Institute of Educational Studies and Research (INEP), according to the guidelines of the National Commission for Higher Education Assessment (CONAES) (INEP 2015).

The Own Assessment Commissions (CPAs in the Portuguese acronym) were conceived as a strategy of articulation of the different subjects and sectors of the HEI for diagnosis and definition of priorities. However, in most universities, there is shrinkage of the proposal and the predominance of a bureaucratic role, compiling data for protocol and mandatory submission to the MEC/INEP. Despite this situation, it is possible to observe more consistent institutional responses in higher education institutions when the work processes of the CPAs are properly supported by university management and anchored in effective participatory processes (Peixoto 2009).

Despite some efforts in universities, with a robust assessment architecture and a gigantic production of information, based on a methodology anchored in individual instruments and, to a lesser extent, collective spaces for reflection and proposition, the quota law and affirmative action policies are neither configured as effective agendas for the institutional evaluation processes developed by HEIs nor as a concern of higher education evaluation studies, especially institutional assessment at this level of education. Although affirmative actions and quotas are understood as an important agenda, they are seen as the responsibility of others, regardless of the existence of bodies explicitly promoting affirmative action or those linked to student support or ethnic–racial issues in universities. The National Higher Education Assessment System (SINAES) is not oriented to favor the institutionalization of affirmative action in universities, often treated as "one more theme" in the field of so-called diversity issues.

There is even a lack of knowledge on the part of certain members of the CPAs about the content of the quota law and its implications for universities. As a result, the interviews carried out within the framework of the research often constituted an informative moment about the law and stimulated the approach to the agenda and dialogue with other sectors and subjects of the university who are on the front line of the debate on affirmative action.

The analysis of the interviews and CPAs' reports suggested that the more decentralized and participatory the institutional evaluation process, with more spaces for listening and collective appropriation and reflection on evaluation results, the greater the possibilities for issues relating to the implementation of the quota law and other types of affirmative actions.

In some reports, the discussion of affirmative action policies, quotas, and ethnic–racial issues are addressed as part of the institutional development project. In others, it is possible to see the almost complete absence of the treatment of affirmative actions in the evaluation documents. There are also cases in which there is data that could be used to improve affirmative action policies, but which lack analysis, debate, and consequences.

This investigation revealed that the field of studies about assessment processes in higher education, especially institutional evaluation, has little dialogue with the effervescence of experiences, demands, and provocations resulting from the advancement of affirmative actions in universities in recent decades, in a context marked in recent years by deep budget cuts and systematic attacks on the autonomy of universities. And that the assessment processes have contributed little to the implementation of the quota law and the improvement and strengthening of affirmative actions and policies committed to racial equality. The indicators provided for in SINAES are incipient and do not favor the evaluation of the implementation of affirmative action experiences developed by universities.

## 4. Conclusions

After presenting the main highlights of the results of the research, in this section, we develop a reflection on the main outcomes of affirmative action in Brazilian public universities and what can be expected on this matter in the coming years.

As has been presented earlier in this paper, the ten years of the quota legislation for federal universities have extensively contributed to the increase of numeric representation of black, brown, indigenous, and low-income students between 2013 and 2022.

This change in the demographic representation of students cannot be seen as the end of the road when it comes to the inclusion and transformation of Brazilian federal universities. Otherwise, as stated by many interviewees in this research, this can be considered the first step for an effective transformation of public HEIs in Brazil.

The second step, broadly addressed in the research, is the need for strong student support policies that can effectively promote student success, with a special focus on quota students. On this matter, the research results show the need for increased public resources, better-designed policies, and greater awareness by different stakeholders in HEIs about the importance of these policies, and their role and responsibilities in the implementation of these actions. The research recommends an expansion of retention policies in place, combining financial aid, pedagogical support, and other strategies to increase the sense of belonging of quota students, expanding their participation in all aspects of university life. One of the components of this expansion is the strengthening of the National Student Assistance Program (PNAES), transforming it into a federal law, with sufficient and stable resources for the implementation of student support policies. These results confirm previous studies carried out in Brazil that highlighted the importance of student support measures to provide significant results related to affirmative action for students from underrepresented groups (Jesus 2019; Prado 2021; D. B. R. Santos 2009; among others).

Affirmative action has effectively contributed to democratizing access to public universities and has created pressure for a profound transformation in the references, meanings, and priorities of universities (research agenda, curricula, changes in procedures, and institutional cultures). However, the institutionalization of these policies in the universities as a whole is still extremely limited, understood in general as the responsibility of bodies and institutions linked to student support or the agenda for the promotion of ethnic and racial equality. There are processes stuck at the intermediate level of universities.

Racial quotas have been decisive for the access of black and indigenous poorer students, demonstrating that only social quotas do not account for democratizing the access of this population to higher education. The presence of blacks, browns, and indigenous students has become more visible in the most selective careers, where, before quotas, they were hardly seen.

Below, we present a set of recommendations aimed at reorienting Sinaes (National Higher Education Assessment) and, specifically, the institutional assessment of higher education from the perspective of greater institutionalization of affirmative action in universities, as part of a system of permanent and participatory monitoring and assessment of the implementation of the quota law.

On the other hand, the research identified the need for better monitoring and evaluation of the quota system, with greater data availability and regular assessment reports to monitor the advancements of the policy, not only in numeric terms but also in relation to the integration of quota students in the universities. This requires the development of measures such as training programs for coordinators, faculty, and administrative staff; there is still a lot of misunderstanding about the quota law, affirmative action policies, and their implications. Universities should integrate the promotion of ethnic–racial equity in their agenda in all institutional dimensions (teaching, research, extension, management), with strong investment in training processes, planning, and monitoring the implementation of actions that confront institutional racism.

The research conclusions propose that the Sinaes (National Higher Education Assessment) is reoriented as part of the monitoring system of the quota law and affirmative action policies, with the inclusion of dimension and review of institutional assessment indicators, in the perspective of strengthening participatory processes, with strong listening to students and other subjects of the university community and promotion of an anti-racist, and non-discriminatory education.

As part of these institutional strategies, it is recommended that the universities articulate affirmative action policies and policies to promote diversity and anti-racist and anti-discriminatory education, including full implementation of the National Curricular Guidelines for the education of ethnic–racial relations and for the teaching of Afro-Brazilian and African history and culture, which establish that higher education institutions include the education of ethnic–racial relations in the contents of subjects and curricular activities.

Another aspect related to this broader approach to affirmative action includes the demand for implementing what is called "epistemic quotas" (curricular diversification in the perspective of Afro-Brazilian, African, and Indigenous matrices, among others) in the universities' careers, in the different areas of knowledge.

The research results also recommend the creation of a National Program for the Dissemination of the quota law and Affirmative Action Policies in High School, with greater articulation between public universities and the state networks of high schools, aiming at greater dissemination of affirmative actions and policies for access and retention of public school students, lower income, black, brown, and indigenous students and people with disabilities in public universities.

The presented analysis shows some prospects. First of all, indicates the need for continuity and strengthening of the reserve of vacancies in access to federal higher education institutions. The research results indicate that Law No. 12,711 should remain in force until its objectives are fully achieved. Its results should continue to be monitored, with an expansion of the dimensions analyzed.

This brings us to an important update in relation to the continuity of the quota legislation in Brazilian federal universities. With the change in the Brazilian federal government in early 2023, and the inauguration of the third term of President Luiz Inacio Lula da Silva, the new congress legislature which also took place at the beginning of 2023 has discussed and approved a new edition of the quota legislation, expanding these measures in time and also modifying some of its measures, in order to expand the effects of the law.

The new quota law, now named Law 14,723, issued on 14 November 2023, brings some important changes made to the previous law, including changes in relation to the program's audience, and to the changes in relation to implementation mechanisms and innovations.

With regard to the target audience of the affirmative action program, the main change concerns the inclusion of *quilombola* students (people from black rural communities of former enslaved Afro-Brazilians) among the benefited groups, alongside black, brown, and indigenous students, lower-income students from public schools, and those with disabilities. This measure meets a demand for recognition of the ethnic and cultural specificity of this group, which is still little considered in targeted policies. This measure is incredibly important given that for the first time, the country collected data on the quilombola population in a demographic census, enabling better knowledge of this group.

In regard to the target audience of the affirmative action program, the new law redefined the income cutoff, reducing the maximum income limit for qualification from 1.5 to 1 minimum wage per capita. In doing so, legislators were aware of the fact already identified in several studies that point to the need for greater focus on the poorest students, so that the policy actually contributes to reducing socioeconomic inequalities in access.

In relation to implementation mechanisms, the most notable advance of the new law in comparison to the previous one is the availability of widely competitive vacancies for all students competing in the Unified Selection System. This measure means that candidates benefiting from affirmative action compete for reserved spots only if they are not approved in the general selection. As several researchers and activists in this field have emphasized, this measure allows quotas to be "a floor and not a ceiling" (Heringer and Carreira 2022; Bó and Senkevics 2023), expanding the entry possibilities for students benefiting from affirmative action. This group will also have priority in the allocation of spots not filled by other methods.

Finally, we highlight the item we call innovation, which concerns two specific aspects that have been debated by experts and also in federal higher education institutions. The first refers to the anticipated adoption of affirmative action policies in graduate programs. This measure formalizes the recommendation that the Ministry of Education and Culture made in 2016, guiding the adoption of these programs and integrating them with the set of affirmative actions in higher education.

The other measure provides quota students who are in a vulnerable situation at the time of entering higher education a priority when it comes to receiving assistance. Although we know that this and other measures provided for in the law are subject to regulation, we highlight the importance of this guidance that aims to provide groups benefiting from entering higher education with the necessary support to remain at the university. The importance of permanence conditions being perceived as part of affirmative action is widely recognized in several studies, guaranteeing special attention to students from these specific groups.

In summary, we can say that law 14,723, by promoting changes to law 12,711, improves mechanisms and also innovates in specific aspects, including new benefiting groups and new monitoring mechanisms.

It is worth mentioning that several of these measures adopted in the new law, regarding quilombola students, income cutoff, changes in the selection process, and affirmative action in graduate programs have been proposed by the research summarized here when it was released in 2022.

After the first ten years of affirmative action in federal universities, our campuses are more similar to our society, with black students, brown, indigenous and white, poor and rich, people with disabilities, living together, facing conflicts, disputing meanings, exchanging experiences, and promoting learning. After ten years, these results indicate that the experience of affirmative action in Brazilian public higher education should continue to focus on the income and race of students to continue the process of redistribution of opportunities and promotion of representativeness in Brazilian federal universities in the perspective of racial and social justice and the decolonization of our universities.

The studies carried out show that much has been achieved, but there is still a need to improve, protect, and strengthen these policies in the coming years.

**Funding:** This research was funded by Open Society Foundations, grant number OR2020-76922.

**Institutional Review Board Statement:** The study was conducted in accordance with the Declaration of Helsinki, and approved by the UFRJ CFCH Ethics Committee (protocol code: CAAE 56179021.5.0000.5582, approved on 30 April 2022).

**Informed Consent Statement:** Informed consent was obtained from all subjects involved in the study.

**Data Availability Statement:** Publicly available datasets were analyzed in this study. This data can be found here: https://drive.google.com/drive/folders/1qRR1iy9ZiM5SR8w1c7DhFE8BEwZx8 2ir?usp=drive_link (accessed on 29 October 2023). The qualitative date presented in this study are available on request from the corresponding author. This data is not publicly available due to ethical reasons, to preserve anonymity of participants.

**Conflicts of Interest:** The author declares no conflicts of interest. The funders had no role in the design of the study; in the collection, analyses, or interpretation of data; in the writing of the manuscript; or in the decision to publish the results.

## Notes

[1] This legislation is a constitutional amendment approved by Brazilian congress that establishes a maximum public spending in several areas of public policies, shrinking the coverage of several social policies in the country.

[2] Although the author and the research team recognize the importance of gender inequalities in higher education, the research presented here chose to focus only on racial and income inequalities, because these were the focus of the quota legislation established in Brazil in 2012. There are horizontal inequalities in the education attainment of women and men, in terms of the career paths of women and men, but in the case of Brazil, women have more years of education and access to higher education in a larger proportion than men.

[3] These committees have been created in several universities to confirm the self-declaration of color/race of black and brown candidates after several denounces from black movement organizations that students who would be socially recognized as whites were declaring themselves black or brown to access the quotas (dos Santos 2021).

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
