# Peer review of "Affirmative Action Policies in Higher Education in Brazil: Outcomes and Future Challenges"

_socsci, doi:10.3390/socsci13030132_

Round 1
Reviewer 1 Report
Comments and Suggestions for Authors
Please see the attached file.

Reviewer 2 Report
Comments and Suggestions for Authors
The research focus on an important and under researched topic, affirmative action in HE in Brazil.
The strength of the article is the rich data collection. Despite this the authors could have developed a better multimethod section where the different data sets could have been discussed more closely; what was the purpose and meaning of collecting such different data sets? How do they relate / not relate? What contradictions and complexities were found using different data? how did the researchers handle the different ontological and epistemological standpoints originating from using quantitative large-scale register data with interviews which focus on experiences from marginalised students? Moreover, what was the purpose of the literature review; How can the meaning of published research contribute to the understanding of lived experiences of marginalised studentgroups and affirmative action policy? I think that the researchers can make better use of their collected data. They need to develop a better methodology sectio so it become clear to the reader why they have decided that they need all this material and more importantly, demonstrate how these data can be analysed jointly and the main outcomes.
The second strength of the article is its clear focus on affirmative action policy. This approach can be further strengthened by referencing other research and by developing a theoretical framework for analysing affirmative action; in other words affirmative action is more than policy or practice - it is a research field with its own theoretical development. As it stands now, the article is under - theorising affirmative action. The authors can cite other research and build a theoretical framework for their analysis. They can draw from critical diversity studies :
Essed, P., & Goldberg, T. (2010). Cloning cultures: The social injustices of sameness. Ethnic and Racial Studies, 25(6), 1066–1108.
Ahmed, S. (2012). On being included: Racism and diversity in institutional life. Duke University Press.
Ahmed, S., Hunter, S., Kilic, S., Swan, E., & Turner, L. (2006). Race, diversity and leadership in the learning and skills sector. Final report, November 2006. https://www.researchgate.net/ publication/242469110_Race_Diversity_and_Leadership_in_the_Learning_and_Skills_Sector. Assessed 8 Feb 2020.
Decolonial approaches to affirmative action research:
Bhambra, G. K., Gebrial, D., & Nişancıoğlu, K. (2018). Introduction. In G. K. Bhambra, D. Gebrial, & K. Nişancıoğlu (Eds.), Decolonising the university? Pluto Press.
Gebrial, D. (2018). Rhodes must fall: Oxford and movements for change. In G. K. Bhambra, D. Gebrial, & K. Nişancıoğlu (Eds.), Decolonising the university. Pluto Press.
Icaza, R., & Vázquez, R. (2018). Diversity or decolonization? Researching diversity at the
University of Amsterdam. In G. K. Bhambra, D. Gebrial, & K. Nişancıoğlu (Eds.),
Decolonising the university. Pluto Press.
Mählck P (2021) Equality, Diversity and Inclusion: How Can Diversity Practice Challenge Racism, Sexism and White Privilege in the Globalised Academy? Cpt 4. In (eds) Carlsson M et al. Gender and Education in Politics, Policy and Practice. Cham, Springer. ISBN 978-3-030-80901-0 *
The third comment refers to gender and gender equality. What was the gender distribution among the disadvantage student groups? It is really a pity that the authors have not done an intersectional analysis of gender and race as it is well known that universitas not only are colonial and racialised institutions but also highly gendered.
Finally, the concluding section needs to be elaborated and discussed beyond simply outlining a list of results. How do the result relate, support, contradict previous national and international research?
Good Luck !
Round 2
Reviewer 1 Report
Comments and Suggestions for Authors
Thank you for following my recommendations for revisions. The presentation of the data and the the new conclusions clearly helps to communicate the importance of your research findings for further research on the topic.
Author Response
Thank you for the sugestions and comments.
Reviewer 2 Report
Comments and Suggestions for Authors
Manuscript is now ready for publication
Author Response

(The authors gave the same response as above.)
